# KNOWLEDGE GUIDED GEOMETRIC EDITING FOR UNSUPERVISED DRUG DESIGN

## ABSTRACT

Deep learning models have been widely used in automatic drug design. Current deep approaches always represent and generate candidate molecules as a 1D string or a 2D graph, which rely on large measurement data from lab experiments for training. However, many disease targets in particular newly discovered ones do not have such data available. In this paper, we propose GEKO, which incorporates physicochemical knowledge into deep models, leading to unsupervised drug design. Specifically, GEKO directly models drug molecules in the geometric (3D) space and performs geometric editing with the knowledge guidance by self-training and simulated annealing in a purely training data free fashion. Our experimental results demonstrate that GEKO outperforms baselines on all 12 targets with and without prior drug-target measurement data.

## 1 INTRODUCTION

Finding a drug molecule that can cure specific diseases is a major and significant challenge in the field of medicine. Traditional drug design methods usually use a screening approach (Bajorath, 2002), such as molecular docking(Ferreira et al., 2015; Meng et al., 2011), to filter molecules from a large database, which cannot generate novel molecules and heavily relies on expert experience for further modification. In recent years, the surge of deep learning has benefited the field of automatic drug design. Intuitively, automatic drug design can be modeled as a conditional generation problem, namely generating a ligand (i.e., drug molecular) from a disease-related target (usually a protein pocket). Currently, majority of the deep learning-based drug design methods express molecules as a SMILES (Weininger, 1988) string (1D) or a graph (2D) , which encourages the application of sequence-based (Gómez-Bombarelli et al., 2018; Kang & Cho, 2018; Segler et al., 2018) and graph-based (Seff et al., 2019; Liu et al., 2018; Ma et al., 2018; Simonovsky & Komodakis, 2018; Samanta et al., 2019; De Cao & Kipf, 2018) models in drug design.

However, deep neural networks that model molecules in 1D/2D formats requires massive experimental activity data to learn relations between low dimensional features and their activities against specific protein targets. This is not only expensive but also limits these models' application scope into proteins with rich activity data. For example, Jin et al. (2018b) formulate the drug design problem as translating one molecule to another with better properties supervised with *labeled molecular pair data*. Similarly, Xie et al. (2021) and Jin et al. (2020) use activity predictors trained from *target-specific activity data* to guide property optimization via Markov chain Monte Carlo and reinforcement learning respectively.

In fact, we can leverage *physicochemical knowledge* to estimate the bio-activities without any labeled data given 3D structural information. In general, a drug molecule exhibits its activity by forming a 3D complex with a disease-related target. The tighter they bind, the more effective is the drug. The tightness can be measured by the minimum binding energy between these two chemical compounds and we have plenty of existing molecular docking software to estimate this energy. However, the searching and optimization for the minimum energy is a computational expensive process, which makes it non-trivial to efficiently extract the knowledge in docking and apply it to automatic drug design.

In this paper, we propose **G**ometric **E**diting under **KnO**wledge guidance (GEKO) to achieve unsupervised drug design. We utilize the knowledge from molecular docking, drug-likeness and synthesizability as these knowledge is broadly applicable for all targets and offers great resources for guid-

ing drug design. Inspired by the knowledge form and a classical medicinal concept called structure activity relation (Guha, 2013)[1], we aim to learn the effects of incremental molecular modifications in a 3D molecular space. In more details, we carefully design a *geometric molecular editing module* to traverse the space by adding or deleting fragments and modeling the *dihedral angles* in between. Then a hierarchical message passing neural network is tailored to capture both atom and fragment level information. We train the network with self-generated data and use simulated annealing to generate molecules under the guidance of knowledge. With a large amount of self-generated data, the model is able to learn the fundamental knowledge for drug design more efficiently and eliminate the dependence on experimental labeled data.

We compare our method with baseline models on 12 targets including those without sufficient experimental data for developing data-driven models. GEKO outperforms other methods on both the 3D molecular space exploration power and capability of generating highly active molecules. The generated 3D molecules show good shape complimentarity with the target surface, which indicates strong interactions between the binding partners.

## 2 RELATED WORKS

**3D Drug Design** Generating 3D molecules is a relatively new area in AI powered drug discovery due to its intrinsic difficulty. Within this field, 3D drug design is a non-trivial task which requires the model to generate new 3D molecules geometrically constrained by the target binding site and also achieve multiple optimized drug-like properties. Comparing to molecular generation in 1D/2D representation, the additional dimension significantly increases the explorable molecular space. Regardless of its significance, limited efforts have been made in this field. Masuda et al. (2020) use a variational autoencoder (VAE) to learn the 3D molecular distribution conditioned on a target structure using `CrossDocked2020` (Francoeur et al., 2020), a 3D ligand-target complex database. Due to the limited amount of experimental measured complex data, many 3D complexes in this dataset are generated using molecular docking. Despite of the large amount of training data, the model still suffers from poor binding affinities of the generated molecules compared to an experimental reference. However, our approach, guided by structural knowledge, can successfully generate molecules with high binding affinity while keeping many other drug-related properties.

**3D Conformation Prediction** Within the scope of 3D molecular generation, 3D conformation prediction is a better studied field comparing to 3D drug design. This task is defined as predicting the 3D structure of a molecule with a given 2D molecular graph, which defines the number of atoms and the connectivity between atoms. These methods can largely be divided into two groups by the way they formulate the 3D generation problem. The first class aims to directly generate the atomic positions or the inter-atomic distance matrix (Mansimov et al., 2019; Simm & Hernandez-Lobato, 2020; Xu et al., 2021; Shi et al., 2021). Methods in this class demonstrate initial success on relative small molecules, such as those in `GEOM-QM9` dataset (Axelrod & Gomez-Bombarelli, 2020). However, their performances drop significantly when the molecular size increases, more specifically on drug-like molecules.

The other class predicts the flexible dihedral angles which is the major cause of conformational flexibility in 3D molecules (Gogineni et al., 2020; Ganea et al., 2021). Modeling dihedral angles shows a more stable performance when the number of rotatable bonds increases (Ganea et al., 2021). Unlike predicting the distance metrics, which tends to overparameterize the degree of freedoms in the molecules, dihedral angles represent the molecular flexibility more straight forward. This kind of approach has been widely used in even larger molecular conformation prediction, namely the proteins. In this work, we integrate this idea into our 3D geometric editing procedure.

**1D and 2D Drug Design** Deep learning methods have been applied to facilitate drug design mainly on the 1D and 2D representations. Drug design is essentially an optimization task, and previously deep models have used various approaches to optimize the properties, such as Bayesian optimization in a latent space (Gómez-Bombarelli et al., 2018; Jin et al., 2018a; Winter et al., 2019), reinforcement learning (De Cao & Kipf, 2018; Popova et al., 2018; You et al., 2018; Popova et al., 2019; Shi et al.,

---

[1]The structure of the drug influences its biological activity. Grasping this relation is the fundamental approach for modifying molecular structure and achieving higher drug activities.

2020), evolutionary and genetic algorithms (Ahn et al., 2020; Jensen, 2019; Devi et al., 2015; Nigam et al., 2020; A Nicolaou et al., 2012) and etc.

More recently, sampling based approach has been used in molecular generation (Fu et al., 2021; Xie et al., 2021). Among them, MARS (Xie et al., 2021) has demonstrated state-of-the-art performance in multi-objective optimization via adaptive Markov chain Monte Carlo sampling. Although this model learns from self-generated data, it still relies on a data-driven predictor for biological activity optimization, which limits it application scope. In addition, directly transferring low dimensional drug design methods to 3D generation is not trivial, as the 3D molecular space is much more complex due to the rotation of dihedral angles.

## 3 THE PROPOSED METHOD

In this section, we address the procedure of GEKO. We first introduce the physical and chemical knowledge used for drug design and the need of 3D geometric editing. Then we describe how to model the automatic drug design as a geometric editing process and its parameterization. Finally, we show how to guide the editing process with these knowledge.

### 3.1 INTEGRATING PHYSICOCHEMICAL KNOWLEDGE IN MOLECULAR GENERATION

We formulate drug design as a generative process with the objective of finding molecules possessing strong bio-activity with respect to the target protein, together with a sufficient level of drug-likeness and synthetic accessibility.

The bio-activity largely depends on the tightness of formed target-ligand complex, which can be measured by the minimum binding energy between two chemical compounds. In particular, GEKO proposes to use **Autodock Vina** (Trott & Olson, 2010; Eberhardt et al., 2021) (or Vina for short), one of the state-of-the-art molecular docking softwares, to estimate this energy.

Drug-likeness is measured by **QED** (Bickerton et al., 2012), which combines molecular weight, number of hydrogen bond donors, molecular surface area and many other drug-like properties. Synthetic accessibility is quantified by **SAscore** (Ertl & Schuffenhauer, 2009) based on the fragment composition and molecular complexity[2]. Combining three metrics, we then have the objective function for molecule $x$:

$$\mathcal{J}(x) = \text{VINA}_{\min}(x) + \alpha \log \text{QED}(x) + \beta \log \text{SASCORE}(x). \tag{1}$$

Though QED and SAscore can be efficiently computed with the 2D structure of molecule, we need extra design to make Vina feasible to use. Specifically, Vina estimates the minimum binding energy by a stochastical global search in 3D conformational space together with local energy optimization (the calculated energy as $\text{Vina}_{\text{dock}}$), which is time-consuming and cannot provide enough estimations in a reasonable amount of time.

In fact, we can get rid of the global search of Vina, which makes it 100x faster, by directly generating in 3D molecular space. The model can learn to generate desired 3D poses so that local energy minimization (the calculated energy as $\text{Vina}_{\min}$) is adequate for energy estimation. Details will be addressed in the following subsections.

### 3.2 TRAVERSING THE 3D MOLECULAR SPACE BY GEOMETRIC EDITING

A molecule's 3D structure can be represented as a composition of rigid 3D fragments linked by bonds with flexible dihedral angles. These angles are formed between two intersecting planes, and in chemistry it is defined using four consecutive atoms and related to the rotation of one bond as shown in Figure 1 *top left*. They are the major cause of conformational flexibility in 3D molecular structures. Inspired by this, we propose to first construct a rigid fragment library, and then edit molecules by either selecting a fragment in the library to add and determining the dihedral angle between the added fragment and the original molecule, or selecting a rigid fragment in the molecule to delete.

---

[2]Note that we re-scale the SAscore to [0, 1].

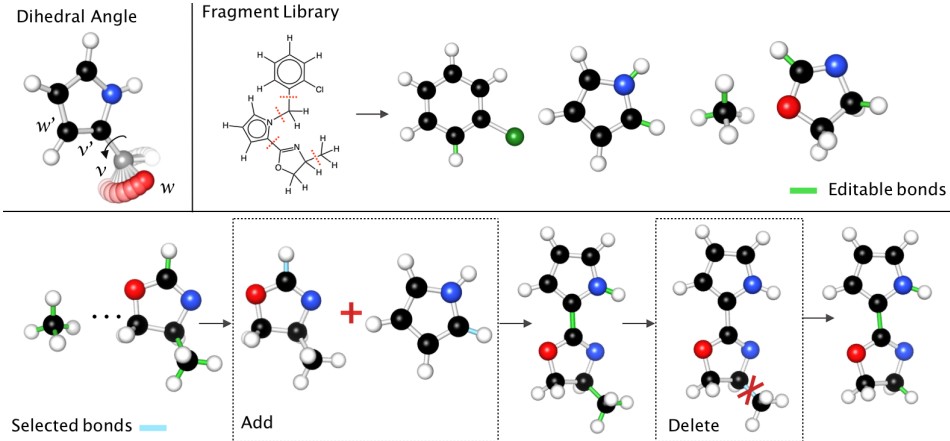

Figure 1: Geometric Editing for 3D Molecular Generation

**Rigid Fragment Library**    Single bond rotation is the major cause of dihedral angle changes, as single bond is less constrained and has lower rotational energy barrier comparing to double bonds and bonds in ring structures. Therefore, we construct a rigid fragment library by breaking single bonds in 2D molecules from the ChEMBL database (Gaulton et al., 2017), which is a manually curated database of bioactive molecules with drug-like properties. As shown in Figure 1 *top right*, the broken bond for generating fragments are labeled as an editable site for molecular construction and a hydrogen atom is added to maintain the original valency[3]. The same 2D fragment with different editable sites are merged as one member in the library and the editable sites are combined. The 3D conformation of the resulted fragments are generated using RDKit (Landrum, 2021). As rotational flexibility is constrained inside fragment, each 2D fragment only corresponds to one or a few valid 3D conformations and each conformation is considered as a separate member in the fragment library.

Using rigid fragments as building blocks can enhance the validity of generated molecular structures and largely reduce the space of possible conformations by taking advantage of existing knowledge.

**Operations of Geometric Editing**    GEKO places an initial seed molecule within a target binding site, and builds 3D molecular structures by adding or deleting rigid fragments iteratively as shown in Figure 1 *bottom* panel. In order to achieve adding and deleting operations separately, the editable sites are categorized as addable sites, which correspond to the editable sites with hydrogen atoms attached, and deletable sites which correspond to the the ones with non-hydrogen atoms attached. In adding operations, the model firstly selects an addable bond in the original molecule. Then it chooses a 3D fragment from the fragment library and an addable bond in the fragment to attach to the original molecule. Lastly, it decides the dihedral angle between the two connected components. The hydrogen atoms in the selected addable bonds are removed resulting in two partial bonds, which are then connected and rotated to the predicted dihedral angle. In deleting operations, the model chooses an deletable bond to break. A hydrogen atom is added to the broken bond to maintain the correct valency.

**Parameterization with Hierarchical MPNNs**    To better model the fragment-based molecular structure, we propose to use hierarchical message passing neural networks (HMPNNs) to predict the probability distributions of editing operations mentioned above. HMPNN is a two-level message passing neural network (Gilmer et al., 2017) that captures the interactions of both atoms and fragments. It takes a molecule $x$ as input, which is represented as a graph $g = (\boldsymbol{A}, \boldsymbol{f}^{\text{node}}, \boldsymbol{f}^{\text{edge}})$ with $\boldsymbol{A}$ as the adjacency matrix, $\boldsymbol{f}^{\text{node}}$ and $\boldsymbol{f}^{\text{edge}}$ as feature vectors. We pass the graph and feature vectors through the first MPNN to obtain atom-level representations:

$$\boldsymbol{h}_u^{\text{node}} = \text{MPNN}_1(g)_u \in \mathbb{R}^d \tag{2}$$

---

[3]If a fragment only contains one atom, it is fused with its connecting fragment.

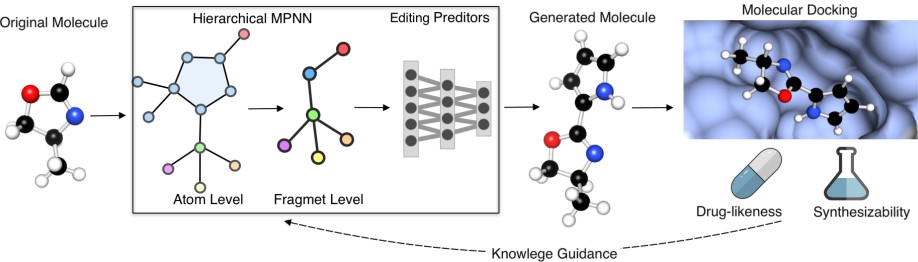

Figure 2: Proposal Architecture for Editing Operation Prediction.

where $\boldsymbol{h}_u^{\text{node}}$ is the hidden representation of node $u$. Note that atoms separated by rotatable single bonds are considered as belonging to different fragments. We then regard each fragment as a single node, which induces a new fragment-level adjacency matrix $\boldsymbol{A}'$, and obtain the fragment embedding $\boldsymbol{z}^{\text{node}}$ by aggregating features of atoms that belong to it using mean pooling. We preserve those edges between fragments with feature vectors $\boldsymbol{z}^{\text{edge}}$ initialized by fragment embeddings.

$$\boldsymbol{z}_i^{\text{node}} = \text{MEANPOOL}_{u \in V_i}(\boldsymbol{h}_u^{\text{node}}) \in \mathbb{R}^d \tag{3}$$

$$\boldsymbol{z}_{j,k}^{\text{edge}} = \boldsymbol{A}_1 \cdot \text{CONCAT}(\boldsymbol{z}_j^{\text{node}}, \boldsymbol{z}_k^{\text{node}}) + \boldsymbol{b}_1 \in \mathbb{R}^d \tag{4}$$

where $V_i$ is the set of atoms in fragment $i$. The new graph $g' = (\boldsymbol{A}', \boldsymbol{z}^{\text{node}}, \boldsymbol{z}^{\text{edge}})$ is then passed to another MPNN to obtain fragment-level representations.

$$\boldsymbol{o}_i^{\text{node}} = \text{MPNN}_2(g')_i \in \mathbb{R}^d \tag{5}$$

$$\boldsymbol{o}_{j,k}^{\text{edge}} = \boldsymbol{A}_2 \cdot \text{CONCAT}(\boldsymbol{o}_j^{\text{node}}, \boldsymbol{o}_k^{\text{node}}) + \boldsymbol{b}_2 \in \mathbb{R}^d \tag{6}$$

With HMPNN as a representation extractor, we can predict the editing operations mentioned before. Let $x_{\text{skel}}$ be the skeleton molecule to be edited. The edge to edit is determined as follows:

$$\boldsymbol{h}^{\text{node}}, \boldsymbol{o}^{\text{node}}, \boldsymbol{o}^{\text{edge}} = \text{HMPNN}_1(x_{\text{skel}}) \tag{7}$$

$$\text{score}_{j,k} = \text{MLP}_1(\boldsymbol{o}_{j,k}^{\text{edge}}) \in \mathbb{R} \tag{8}$$

$$p_{\text{add}}(r|x_{\text{skel}}) = \text{softmax}(\{\text{score}_{j,k}\}_{(j,k) \in E_a}) \tag{9}$$

$$p_{\text{delete}}(r|x_{\text{skel}}) = \text{softmax}(\{\text{score}_{j,k}\}_{(j,k) \in E_d}) \tag{10}$$

where $E_a$ and $E_d$ are disjoint sets of editable edges on $x_{\text{skel}}$ for addition and deletion respectively. During generation, we sample a directed edge $r = (u_{\text{skel}}, v_{\text{skel}})$ from $\frac{1}{2}p_{\text{add}} + \frac{1}{2}p_{\text{delete}}$. For deletion, we just replace the deleted fragment on $u_{\text{skel}}$ side with a hydrogen. For addition, we predict the fragment $x_{\text{frag}}$ to add also on $u_{\text{skel}}$ side:

$$\text{score}_{x_{\text{frag}}} = \text{MLP}_2(\boldsymbol{o}_r^{\text{edge}})_{x_{\text{frag}}} \in \mathbb{R} \tag{11}$$

$$p_{\text{fragment}}(x_{\text{frag}}|x_{\text{skel}}, r) = \text{softmax}(\{\text{score}_{x_{\text{frag}}}\}_{x_{\text{frag}} \in H}) \tag{12}$$

where $H$ is the rigid library. Then we sample a fragment $x_{\text{frag}}$ from $p_{\text{fragment}}$. The attachable bond on the fragment is then determined jointly by $x_{\text{skel}}$ and $x_{\text{frag}}$:

$$\boldsymbol{h}^{\text{frag-node}}, \boldsymbol{o}^{\text{frag-node}}, \boldsymbol{o}^{\text{frag-edge}} = \text{HMPNN}_2(x_{\text{frag}}) \tag{13}$$

$$\bar{\boldsymbol{o}}^{\text{node}} = \text{MEANPOOL}(\boldsymbol{o}^{\text{node}}) \tag{14}$$

$$\text{score}_{j,k}^{\text{frag}} = \text{MLP}_3(\text{CONCAT}(\bar{\boldsymbol{o}}^{\text{node}}, \boldsymbol{o}_{j,k}^{\text{frag-edge}})) \in \mathbb{R} \tag{15}$$

$$p_{\text{attach}}(a|x_{\text{skel}}, r, x_{\text{frag}}) = \text{softmax}(\{\text{score}_{j,k}\}_{(j,k) \in E_a^{\text{frag}}}) \tag{16}$$

where $E_a^{\text{frag}}$ is the set of attachable bonds on fragment $x_{\text{frag}}$. Then we sample a bond $a = (u_{\text{frag}}, v_{\text{frag}})$ to attach. Finally, we gather the features of four atoms, $v_{\text{skel}}$, $w_{\text{skel}}$, $v_{\text{frag}}$, $w_{\text{frag}}$, around the new bond[4]and determine the dihedral angle:

$$h^{\text{angle}} = \text{CONCAT}(h_{v_{\text{skel}}}^{\text{node}}, h_{w_{\text{skel}}}^{\text{node}}, h_{v_{\text{frag}}}^{\text{frag-node}}, h_{w_{\text{frag}}}^{\text{frag-node}}) \tag{17}$$

$$\text{score}_\alpha^{\text{angle}} = \text{MLP}_4(h^{\text{angle}})_\alpha \in \mathbb{R} \tag{18}$$

$$p_{\text{angle}}(\alpha | x_{\text{skel}}, r, x_{\text{frag}}, a) = \text{softmax}(\{\text{score}_\alpha^{\text{angle}}\}_{\alpha \in A}) \tag{19}$$

and sample an angle $\alpha$ from $p_{\text{angle}}$. We then concatenate the skeleton molecule and fragment molecule together by this dihedral angle $\alpha$.

## 3.3 EDITING UNDER KNOWLEDGE GUIDANCE

We use adaptive self-learning to train the prediction network and simulated annealing to sample molecules under the guidance of physiochemical knowledge.

**Adaptive Self-Learning** A dataset $D$ is collected on-the-fly with the editing process. Denote $x$ and $x'$ to be the molecule before and after edit respectively. If the objective score of $x'$ is higher than $x$, molecule pair $(x, x')$ is added to $D$. Direct maximum likelihood estimation (MLE) training on self-generated dataset $D$ provides limited supervision signal. Instead, we train our proposal using weighted maximum likelihood estimation (WMLE) as follows.

$$\arg\max_\theta \frac{1}{|D|} \sum_{(x,x') \in D} \lambda(x', x) \log q_\theta(x'|x) \tag{20}$$

where $q_\theta$ is the network and $\lambda(x', x)$ is a monotonic function indicating the score difference between $x'$ and $x$. Here we choose $\lambda(x', x) = \min\{\mathcal{J}(x') - \mathcal{J}(x), 5\}$. WMLE is expected to offer less biased gradient estimation with respect to the objective function.

**Simulated Annealing** We use simulated annealing (Laarhoven & Aarts, 1987) to approximate the global optimum of the given objective function $\mathcal{J}(x)$ in 3D molecular space $X$. Starting from an initial molecule $x_0$, we use the prediction network to propose some neighboring molecule $x'$ of the current one $x_t$ at each time step $t$. The proposed molecule can either be accepted $x_{t+1} = x'$ or rejected $x_{t+1} = x_t$ as determined by an acceptance probability $\mathcal{A}(x', x_t) = \min\{1, \exp(\frac{\mathcal{J}(x') - \mathcal{J}(x_t)}{T})\}$, where $T$ is the annealing temperature controlling how greedy the process is.

## 4 RESULTS AND DISCUSSIONS

### 4.1 EXPERIMENTS

We evaluate our method on drug design tasks using two sets of biological targets:

- **Target Set A**: a diverse set of 10 biological targets with PDB IDs of 1FKG, 2RD6, 3H7W, 3P0P, 3VRJ, 4CG9, 4OQ3, 4PS7, 5E19 and 5MKU used by Masuda et al. (2020). Targets in this set do not have available data-driven activity predictors. Hence, 2D molecule generation methods cannot be applied to these cases.
- **Target Set B**: a set of 2 targets, namely JNK3 and GSK3$\beta$ (PDB IDs: 3FI2 and 4J71) Well-performing ligand-based scoring functions have been developed for these targets using experimental measured data (Jin et al., 2020).

We compare our method with 3D molecular drug design models using both Target Set A and B since these models can be generalized to all targets. A comparison with 2D drug design models is only conducted on Target Set B due to their limited application scope.

---

[4]where $A = \{0, 10, 20, \cdots 350\}$. Note that $w_{\text{skel}}$ is the neighboring atom of $v_{\text{skel}}$ in skeleton and $w_{\text{frag}}$ is that of $v_{\text{frag}}$ in the added fragment as shown in Figure 1 *top left*

**Baselines** We compare GEKO with the following baselines. **liGAN** (Masuda et al., 2020) is a drug design method that can generated 3D molecules conditioned on a target binding site. It represents molecules as atomic density grids, and uses conditional variational autoencoders in conjunction with a GAN loss to learn the distribution in existing ligand-target complexes. The atom types and 3D coordinates are determined by optimizing the fitness to the generated atomic density grids. **JT-VAE** (Jin et al., 2018a) represents 2D molecule as junction trees and uses variational autoencoders to generate new molecules. It employs Bayesian optimization to achieve property optimization. **RationaleRL** (Jin et al., 2020) constructs new 2D molecules by extracting and combining rationales for desired properties. **GA+D** (Nigam et al., 2020) uses genetic algorithm (GA) to search for molecules with desired properties via explicit editing. It equips GA with a deep neural network penalty to promote its exploratory behaviour and increase the diversity of the generated molecules. **GraphAF** (Shi et al., 2020) a flow-based autoregressive model for molecular graph generation. **MolEvol** (Chen et al., 2021) uses an Expectation- Maximization (EM) like explainable evolutionary approach to optimize molecular properties. Following the method of RationaleRL, it also extracts rationales and conducts a molecular graph completion. **MolDQN** (Zhou et al., 2019) is a reinforcement learning-based molecular optimization method. **MARS** (Xie et al., 2021) is a state-of-the-art method, which uses adaptive Markov chain Monte Carlo sampling to generate 2D molecules by explicit editing.

**Evaluations** We evaluate the models' performance from two aspects: 1. the overall quality of the generated molecules and 2. the capability of generating highly active molecules.

For the overall quality, the generated molecules should cover a diverse chemical space and also not overlap with existing actives. In addition, the generated molecules should fulfill the basic requirements for potential drug molecules. More specifically, we use the following metrics to measure the overall quality: **Uniqueness (Uniq)** is the percentage of unique molecules among all generated ones. **Novelty (Nov)** measures the percentage of generated molecules with similarity less than 0.4 compared to its nearest neighbor in the existing actives for the target of interest (Olivecrona et al., 2017). Similarity is calculated based on pairwise Tanimoto similarity using Morgan fingerprints (Rogers & Hahn, 2010). **Diversity (Div)** measures the internal diversity of the generated molecules, which is formularized as $\frac{2}{n(n-1)} \sum_{x \neq x' \in G} 1 - sim(x, x')$. Morgan fingerprint with radius of 3 is used to calculate similarity. **Success rate (SR)** calculates the percentage of generated molecules that pass the predefined thresholds for desired properties. In this work, we define a successful molecule to have QED $\geq 0.25$, SAscore $\geq 0.59$, and Vina$_{dock}$ score $\leq -8.18$ kcal/mol. QED and SAscore thresholds are defined as the 10th percentile of approved drugs in DrugCentral (Ursu et al., 2019) and the intuition is to cover majority of real drugs. Vina$_{dock}$ score is a binding energy evaluation using the global energy optimization algorithm in Vina. Its threshold corresponds to a binding affinity less than 1 $\mu$M, which is a widely used rule of thumb to guarantee a moderate biological activity in medicinal chemistry. For 3D baselines, the generated 3D molecules are directly docked using Vina, while, for 2D baselines, the generated 2D molecules are embedded into 3D conformers using RDKit and then docked using Vina. **Product (Prod)** is the product of the four aforementioned metrics, which serves as a comprehensive evaluation.

To evaluate the models' capability of generating highly active molecules, we analyze the distribution of biological activities using Vina global search reflected by Vina$_{dock}$ score. Unlike QED and SAscore, Vina$_{dock}$ score is the higher the better. We use the **Median Vina$_{dock}$ (Median)** and the average of **Top 10 Vina$_{dock}$ (Top 10)** to quantify the activity distribution.

**Experimental Setup** We generate 1000 molecules for each target and the target 3D structures are downloaded from CrossDocked2020 (Francoeur et al., 2020). 3D rigid fragments are restricted to include no more than 10 heavy atoms and top 1000 most frequent fragments are selected as the fragment library. The generated molecules are restrict to contain no more than 40 heavy atoms. Starting from methane, 5000 Markov chains proceed simultaneously to ensure sufficient data for on-the-fly training. 1000 molecules are randomly selected at the last step for performance evaluation. HMPNN model contains 6 atomic layers and 3 fragment layers. The atomic node features includes atomic number, element type, charge and 3D coordinates, and the atomic edge features include bond type. The hidden layer node embedding has a size of 64. For model training, we use an Adam optimizer (Kingma & Ba, 2015) to update the model's parameters with an initial learning rate of $3 \times 10^{-4}$. The training dataset $D$ is set to have a maximum size of 75000 samples.

Table 1: Comparison of different drug design methods

| Targets | Method | Uniq (%) | SR (%) | Nov (%) | Div | Prod | Median (kcal/mol) | Top 10 (kcal/mol) |
|---------|--------|----------|--------|---------|-----|------|-------------------|-------------------|
| Set A | liGAN | 99.8 ± 0.2 | 1.0 ± 0.9 | 100 ± 0.0 | **0.922 ± 0.001** | 0.01 ± 0.01 | -5.91 ± 0.43 | -8.79 ± 0.71 |
| | GEKO | **100 ± 0.0** | **57.0 ± 13.2** | **100 ± 0.0** | 0.912 ± 0.004 | **0.52 ± 0.12** | **-9.58 ± 1.01** | **-12.23 ± 1.20** |
| Set B | JT-VAE | 96.8 ± 0.2 | 11.3 ± 0.8 | 100 ± 0.0 | 0.910 ± 0.001 | 0.10 ± 0.01 | -8.21 ± 0.30 | -11.38 ± 0.13 |
| | RationaleRL | 98.8 ± 0.7 | 26.7 ± 14.3 | 33.4 ± 2.0 | 0.883 ± 0.010 | 0.07 ± 0.04 | -7.76 ± 0.21 | -9.56 ± 0.23 |
| | GA + D | 40.2 ± 3.7 | 23.7 ± 20.9 | 86.5 ± 13.5 | 0.849 ± 0.012 | 0.05 ± 0.04 | -7.21 ± 1.00 | -9.94 ± 1.04 |
| | GraphAF | 91.0 ± 0.4 | 0.5 ± 0.3 | 100 ± 0.0 | **0.946 ± 0.000** | 0.00 ± 0.00 | -4.51 ± 0.18 | -8.85 ± 0.22 |
| | MolDQN | 56.2 ± 6.1 | 0.0 ± 0.0 | 100.0 ± 0.0 | 0.784 ± 0.043 | 0.00 ± 0.00 | -5.54 ± 0.19 | -6.95 ± 0.13 |
| | MolEvol | 97.8 ± 0.4 | 41.4 ± 17.0 | 62.6 ± 18.4 | 0.737 ± 0.045 | 0.17 ± 0.03 | -8.17 ± 0.19 | -9.26 ± 0.44 |
| | MARS | 84.7 ± 1.3 | 30.3 ± 24.7 | 90.1 ± 0.9 | 0.826 ± 0.002 | 0.19 ± 0.15 | -7.73 ± 0.54 | -9.54 ± 0.57 |
| | liGAN | 99.8 ± 0.1 | 0.2 ± 0.1 | 100 ± 0.0 | 0.923 ± 0.002 | 0.00 ± 0.00 | -5.34 ± 0.00 | -8.03 ± 0.09 |
| | GEKO | **100 ± 0.0** | **57.1 ± 2.1** | **100 ± 0.0** | 0.910 ± 0.001 | **0.52 ± 0.02** | **-9.22 ± 0.14** | **-11.84 ± 0.54** |

## 4.2 RESULTS AND ANALYSIS

We compare GEKO with baseline models on Target Set A and B and the average results over targets in each set are summarized in Table 1. More detailed performance on each target can be found in the appendix. The performance of 2D molecular generation baselines, namely JT-VAE, RationaleRL, GA + D, GraphAF, MolDQN, MolEvol and MARS, is only reported on Target Set B. From the results, it is obvious that GEKO outperforms baseline models on all evaluation metrics except for diversity. In Target Set A, GEKO surpasses liGAN on the Prod score, which is an indicator for the chemical space exploration power. In Target Set B, it surpasses the best baseline MARS by 3-fold on the Prod score. In addition, GEKO achieves a median Vina$_{dock}$ score of -9.58 kcal/mol and -9.22 kcal/mol respectively in Target Set A and B. It is lower than the best-performing baseline JT-VAE by about 1 kcal/mol, which corresponds to about 5-fold increase in the biological activity. In terms of diversity, GEKO obtains a comparable performance.

Among all these baseline methods, liGAN is the only 3D molecular generation method, but its performance is not satisfactory. liGAN uses a 3D ligand-target complex structure database, Cross-Docked2020, as its training set, and employs a VAE to learn the structural distribution of the 3D ligands. However, not all the molecules in this dataset are drug-like or highly active. Therefore, it is not guaranteed that the generated molecules can be better than an experimental measured active molecule especially without a designed optimization procedure in the 3D space.

2D drug design methods, due to their data driven nature, can only be applied to targets with rich experimental data, which makes them unsuitable in many application cases. Also when using physico-chemical evaluation metrics, their performance can not be guaranteed. MARS achieves the best Prod score among all 2D drug design baselines. It demonstrates a moderate performance in both chemical space exploration and capability of generating active molecules. MolEvol has the best success rate among all 2D drug design methods. However, as both MolEvol and RationaleRL extracts substructures from existing actives as rationales for property optimization, they suffer low novelty which hurts their overall performance. JT-VAE achieves good median and top 10 Vina$_{dock}$ scores, but it suffers greatly in the success rate. GA+D uses genetic algorithm to search for molecules with desired properties. GA can easily get trapped in local minima, which leads to the low uniqueness. MolDQN suffers from low uniqueness and success rate. GraphAF has relatively weak performance on Prod score and Vina$_{dock}$ since many of its generated molecules are small and cannot meet the drug requirements.

We visualize the generated molecules and inspect their physical interactions with the corresponding targets. Figure 3 shows an example of GEKO generated molecule for MDM2 target (PDB ID: 4OQ3). The binding site of MDM2 protein features three sub-pockets indicated as the colored region in Figure 3. Most MDM2 inhibitors has three arms to reach into the sub-pockets, and the binding mode of a strong MDM2 inhibitor is shown in Figure 3 *left*. Majority of the successful molecules generated by GEKO also have the three-arm structure and the binding mode of a candidate with highest Vina$_{min}$ score is shown in Figure 3 *middle* and *right*. From the illustration, it is clear that the generated molecule has a good shape complimentarity with the target binding site, which is important for forming strong physical interactions.

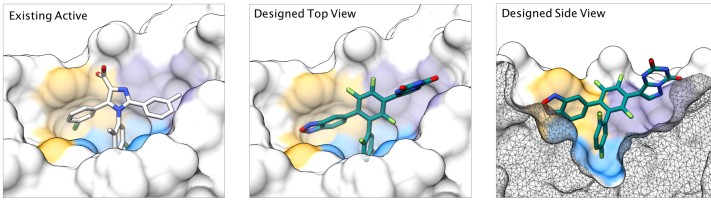

Figure 3: 3D Binding Pose Comparison between Existing Actives and Generated Molecules

## 4.3 ABLATION STUDY

In this section we study the contributions of HMPNN model and weighted maximum likelihood estimation loss in the adaptive proposal using JNK3 target. The comparisons are performed using 1000 sampling chains. Table 2 summarizes the performance of different proposal strategies. **Random** represents a proposal which randomly selects any bonds and fragments to modify the molecular structure. **HMPNN (Random Weight)** represents HMPNN model with random weights. **MPNN** stands for using MPNN to propose editing operations, whereas, **HMPNN** indicates the use of hierarchical MPNN instead. Except for the Random proposal, all other proposals only allows editing on the fragment level as part of our architecture design. In addition, all proposals are equipped with simulated annealing for property optimization. The results indicates the adaptive proposal parameterized using HMPNN is better than the random proposal reflected by the Prod score, median and top 10 Vina$_{dock}$ scores. HMPNN (Random Weight) shows better performance than Random proposal in the Prod score and obtains similar performances on Vina$_{dock}$ scores, which reflects editing on the fragment level is more effective in drug-like space exploration. Comparing to MPNN model, the addition of a fragment layer also leads to a performance increase in Prod score and Top 10 Vina$_{dock}$ and achieves a comparable result in Median Vina$_{dock}$.

Table 2: Comparison of different proposal strategies

| Method | Uniq (%) | SR (%) | Nov (%) | Div | Prod | Median (kcal/mol) | Top 10 (kcal/mol) |
|---|---|---|---|---|---|---|---|
| Random | 100.0 | 31.4 | 100.0 | 0.909 | 0.29 | -8.80 | -12.09 |
| HMPNN (Random Weight) | 100 | 50.0 | 100 | 0.914 | 0.46 | -8.91 | -11.99 |
| MPNN | 100 | 54.1 | 100 | 0.911 | 0.49 | -9.26 | -12.28 |
| HMPNN | 100 | 58.2 | 100 | 0.911 | 0.53 | -9.24 | -12.50 |

In addition, we compare the effects of different loss functions in training the adaptive proposal (Table 3). **WMLE**, on top of the standard **MLE**, takes the score difference $f(x') - f(x)$ into consideration. Using WMLE results in a dramatic improvement on the model's performance. By explicitly formulating the objective score difference in the loss function, the model can more easily learn edits leading to significant improvements than those with moderate boosts.

Table 3: Comparison of different loss functions

| Method | Uniq (%) | SR (%) | Nov (%) | Div | Prod | Median (kcal/mol) | Top 10 (kcal/mol) |
|---|---|---|---|---|---|---|---|
| MLE | 100 | 36.7 | 100 | 0.915 | 0.34 | -8.59 | -11.96 |
| WMLE | 100 | 58.2 | 100 | 0.911 | 0.53 | -9.24 | -12.50 |

## 5 CONCLUSION

In this paper, we propose GEKO, which is a generalizable model to design drug molecules conditioned on a target binding site. We formulate the drug design and optimization task as a step-wise editing process and developed a hierarchical MPNN model to learn the structure-activity relation and facilitate the design process. Our method shows great potential on 3D drug design tasks with a 3-fold performance increase comparing to the best baseline model.

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

## A APPENDIX

Table 4: Comparison of methods on individual target in Target Set A

| Method | Target | Uniq (%) | SR (%) | Nov (%) | Div | Prod | Median (kcal/mol) | Top 10 (kcal/mol) |
|---|---|---|---|---|---|---|---|---|
| liGAN | 3H7W | 99.9 | 0.0 | 100.0 | 0.923 | 0.00 | -5.33 | -7.47 |
| | 5E19 | 99.8 | 1.5 | 100.0 | 0.923 | 0.01 | -6.11 | -9.43 |
| | 2RD6 | 100.0 | 1.0 | 100.0 | 0.921 | 0.01 | -5.78 | -9.11 |
| | 3VRJ | 99.7 | 3.1 | 100.0 | 0.922 | 0.03 | -6.67 | -9.25 |
| | 4OQ3 | 100.0 | 0.0 | 100.0 | 0.923 | 0.00 | -5.20 | -7.57 |
| | 4CG9 | 99.6 | 0.8 | 100.0 | 0.922 | 0.01 | -6.18 | -9.29 |
| | 3P0P | 99.9 | 0.7 | 100.0 | 0.920 | 0.01 | -6.27 | -9.24 |
| | 4PS7 | 99.9 | 0.4 | 100.0 | 0.923 | 0.00 | -5.77 | -8.60 |
| | 5MKU | 99.6 | 1.9 | 100.0 | 0.923 | 0.02 | -6.19 | -9.52 |
| | 1FKG | 100.0 | 0.1 | 100.0 | 0.921 | 0.00 | -5.66 | -8.47 |
| GEKO | 3H7W | 100.0 | 53.4 | 100.0 | 0.914 | 0.49 | -8.52 | -11.05 |
| | 5E19 | 100.0 | 69.7 | 100.0 | 0.916 | 0.64 | -9.60 | -12.61 |
| | 2RD6 | 100.0 | 50.3 | 100.0 | 0.908 | 0.46 | -11.69 | -14.33 |
| | 3VRJ | 100.0 | 67.9 | 100.0 | 0.919 | 0.62 | -9.54 | -12.21 |
| | 4OQ3 | 100.0 | 40.6 | 100.0 | 0.909 | 0.37 | -8.48 | -10.78 |
| | 4CG9 | 100.0 | 51.3 | 100.0 | 0.907 | 0.47 | -9.89 | -12.54 |
| | 3P0P | 100.0 | 67.7 | 100.0 | 0.916 | 0.62 | -10.28 | -13.41 |
| | 4PS7 | 100.0 | 61.6 | 100.0 | 0.907 | 0.56 | -10.37 | -12.92 |
| | 5MKU | 100.0 | 75.6 | 100.0 | 0.916 | 0.69 | -9.36 | -12.28 |
| | 1FKG | 100.0 | 31.8 | 100.0 | 0.911 | 0.29 | -8.10 | -10.12 |

Table 5: Comparison of methods on individual target in Target Set B

| Method | Target | Uniq (%) | SR (%) | Nov (%) | Div | Prod | Median (kcal/mol) | Top 10 (kcal/mol) |
|---|---|---|---|---|---|---|---|---|
| JT-VAE | JNK3 | 96.9 | 10.4 | 100.0 | 0.911 | 0.09 | -7.91 | -11.50 |
| | GSK3$\beta$ | 96.6 | 12.1 | 100.0 | 0.908 | 0.11 | -8.50 | -11.25 |
| RationaleRL | JNK3 | 98.1 | 41.0 | 31.4 | 0.873 | 0.11 | -7.97 | -9.78 |
| | GSK3$\beta$ | 99.4 | 12.4 | 35.3 | 0.893 | 0.04 | -7.55 | -9.33 |
| GA + D | JNK3 | 36.5 | 44.6 | 73.0 | 0.837 | 0.10 | -8.21 | -10.98 |
| | GSK3$\beta$ | 43.9 | 2.7 | 100.0 | 0.860 | 0.01 | -6.21 | -8.90 |
| GraphAF | JNK3 | 90.6 | 0.1 | 100.0 | 0.946 | 0.00 | -4.69 | -8.63 |
| | GSK3$\beta$ | 91.4 | 0.8 | 100.0 | 0.946 | 0.01 | -4.33 | -9.07 |
| MolDQN | JNK3 | 50.2 | 0.0 | 100.0 | 0.827 | 0.00 | -5.35 | -6.82 |
| | GSK3$\beta$ | 62.3 | 0.0 | 100.0 | 0.741 | 0.00 | -5.74 | -7.08 |
| MolEvol | JNK3 | 98.2 | 58.4 | 44.2 | 0.782 | 0.20 | -8.36 | -9.70 |
| | GSK3$\beta$ | 97.4 | 24.4 | 81.1 | 0.692 | 0.13 | -7.98 | -8.82 |
| MARS | JNK3 | 83.4 | 54.9 | 91.0 | 0.825 | 0.34 | -8.27 | -10.12 |
| | GSK3$\beta$ | 85.9 | 5.6 | 89.3 | 0.828 | 0.04 | -7.19 | -8.97 |
| liGAN | JNK3 | 99.8 | 0.1 | 100.0 | 0.921 | 0.00 | -5.34 | -7.94 |
| | GSK3$\beta$ | 99.9 | 0.2 | 100.0 | 0.924 | 0.00 | -5.33 | -8.12 |
| GEKO | JNK3 | 100.0 | 55.0 | 100.0 | 0.911 | 0.50 | -9.36 | -12.38 |
| | GSK3$\beta$ | 100.0 | 59.2 | 100.0 | 0.909 | 0.54 | -9.07 | -11.30 |

We visualize the evolution of adaptive geometric editing proposal under knowledge guidance using UMPA(McInnes et al., 2018) in Figure 4. The illustration in Figure 4.A shows chemical space coverage changes with steps. It indicates a directional shift as the Markov chains proceeding. Figure 4.B shows the objective score changes over steps, and a clear improvement of objective score is observed.

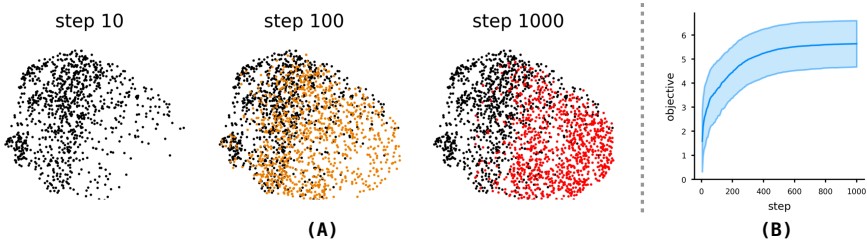

Figure 4: Visualization of Properties Changes Over Steps. A. Chemical Space Coverage. Visualization of chemical space coverage at step 10 (black), step 100 (yellow) and step 1000 (red). B.Objective Score.

