# OpenReview forum: "Knowledge Guided Geometric Editing for Unsupervised Drug Design"
_ICLR.cc/2022/Conference — ICLR 2022 Submitted_

### Official Review · Reviewer_gg4D · 2021-11-01

**Correctness:** 2
**Technical Novelty And Significance:** 2
**Empirical Novelty And Significance:** 2
**Recommendation:** 3
**Confidence:** 4

**Main Review:**


The high level contours of the technical work sound reasonable;
however, by reading the paper I was not convinced that the eventual
deep learning model adds much value to this process. If I take the
authors ablation study by heart, then a random way of picking editable
bonds would still result in a fantastic score, way beyond any of the
scores of previously published chemistry models, even those in the 2D
case. So is the result simply a tuned simulated annealing of fragments
or is there any important learned aspect on picking from the library?

Unfortunately, this paper is not written in a clear fashion and leaves
open questions relating to reproducibility. There is a lack of a clear
exposition of the details with regards to the datasets, benchmarks,
and tons of additional little details peppered throughout the paper:
does the fingerprint used for similarity include chirality and what is
its radius? How large was the resulting rigid fragment library? (At
some point the authors mention including only fragments with less than
10 atoms, but even the single aromatic ring in Figure 1 has 12 atoms
total---did the authors mean heavy atoms when they say atoms?) What
were the training/starting molecules in each case and do those matter
(if not, do the authors start from a methane every time?) Since the
authors seem to have used docking together with the 2D models, why
didn't they also use the 2D generator models on the 3D benchmarks of
table 1? (Incidentally, in Table 1, the largest score in column Div
seems to be the one for GraphAF, not liGAN.) The explanation of the
authors that GraphAF generated small molecules that don't meet the
drug requirements seems interesting; I thought that this model can be
trained to generate arbitrary large molecules---is that not the case?
What targets did the authors evaluate in table 2, 3; and why are the
baseline results different from those in table 1? I could not find out
what the scatterplot in Figure 4 represents---is that some abstract
view of chemical space, and if so what exactly are the x- and y- axes?
What is the deepSAR model in tables 4, 5? (I checked that the authors
don't refer to the paper with the model called deepSAR, but I suspect
that they might have meant GEKO in those tables; additionally, in
table 5, I suppose the authors meant to refer to Target Set B, not A).


**Summary Of The Paper:**

The paper proposes a method that designs molecules by combining
fragments along "editable" (single) bonds. The method uses two graph
neural networks (on atoms and on fragments) to learn to pick editable
bonds. The method also uses simulated annealing to guide the
combination of fragments (and possibly their dihedral angles) to
optimize a weighted score that the authors have found helps their
training. The model is able to propose reasonable molecules in 3D
conformations within a ligand pocket using a docking score for
guidance. The authors perform an ablation study that shows that even
without the deep neural network for selecting the editable bonds,
their sampling methodology performs rather well.


**Summary Of The Review:**


I have concerns about the reproducibility of this work and I find a
lot of details lacking. The English needs some work; some figures and
tables have unexplained or mislabeled data. It is possible that with a
lot of additional work the same idea could become clearer and well
documented, but I am not confident it could make it to ICLR22.

---

> ### Author Response · Authors · 2021-11-20
> **Response to Reviewer gg4D**
>
> We thank Reviewer gg4D for the careful evaluation and in the following paragraph. Following the reviewer's suggestions, we update our paper with more method details. In the  following paragraph, we address the reviewer's concerns.
>
> - Ablation study:
> |Method|Uniq|SR|Nov|Div|Prod|Median|Top 10|
> |---|---|---|---|---|---|---|---|
> |Random|100.0| 31.4 | 100.0 | 0.909 | 0.29 | -8.80 |-12.09|
> |HMPNN (random weight) |100|50.0|100|0.914|0.46|-8.91|-11.99|
> |MPNN|100|54.1|100|0.911|0.49|-9.26|-12.28|
> |HMPNN|100|58.2|100|0.911|0.53|-9.24|-12.50|
>
> The original "random" results reported in Table 2 represent random weights for HMPNN model. Therefore, the editing procedure is still constrained by the architecture of the HMPNN model, which only allows editing on the fragment level. To make the table more clear, we change "random" to "HMPNN (random weight)", and in addition we add a random model to represent randomly selecting bonds and angles to conduct editing on atomic level. The new random proposal shows an obvious performance drop on Prod Score, which indicates that our designed fragment-based editing architecture plays an important role. In addition, all the results reported in Table 2 are implemented in our framework powered by simulated annealing, which also contributes to the optimization of molecular properties.
>
> - Method Details:
> 1. Q: What fingerprint is used for similarity include chirality and what is its radius? \
>    Morgan fingerprint with radius of 3 and chirality is not included.
> 2. Q: How large was the resulting rigid fragment library? \
> We keep the top 1000 frequent fragments in our library.
> 3. Q: What were the training/starting molecules in each case and do those matter (if not, do the authors start from a methane every time?) \
> The results reported always start from a methane.
> 4. Q: Since the authors seem to have used docking together with the 2D models, why didn't they also use the 2D generator models on the 3D benchmarks of table 1? \
> We didn't use molecular docking in 2D models. As we discussed in the "introduction" and "results and discussion" 2D methods are equipped with data-driven activity predictors developed using experimental measured data. These predictors are not generalizable to all proteins. The targets in Target Set A do not have available data-driven activity predictors, which makes them not suitable for 2D models.
> 5. Q: GraphAF generated small molecules that don't meet the drug requirements seems interesting; I thought that this model can be trained to generate arbitrary large molecules is that not the case? \
> GraphAF model has two terminating conditions. One is that the molecule reaches the max size, which is 48 here. The other is that the model generates a special type of edge to signal the termination. In the experiment we keep the objective function of GraphAF the same as other 2D baselines for fair comparison and the observation is that its generated molecules are smaller than other baselines which means the model terminates the generation quickly by the second condition.
> 6. Q: What targets did the authors evaluate in table 2, 3; and why are the baseline results different from those in table 1? \
> We use JNK3 as an example to conduct ablation study and to save computational cost, we run reduce the number of chains to 1000. Table 1 represents the average of all targets and Table 2 and 3 contains results of only one target, therefore the results are different.
> 7. Q: I could not find out what the scatterplot in Figure 4 represents---is that some abstract view of chemical space, and if so what exactly are the x- and y- axes? \
> Figure 4 is a representation of the explored chemical space visualized using UMap. The x- and y- axes are two mapping axes.

---

> > ### Comment · Reviewer_gg4D · 2021-11-23
> > **Thank you for the additional details--some additional minor clarifications**
> >
> > I have a couple of minor additional questions based on your answers.  Importantly, do the original random results include any learned part of the model, or not?  (Do they simply include the architecture, but random weights, or is there any other part of the model that trains on data?)  Why do you think that your original random method beats all past benchmark results in such a dramatic way without any learning?  Is it simply the simulated annealing based on a reasonable choice of editable bonds or is there anything else?
> >
> > With regards to your answers about the method details, as numbered in your reply: 1. why exclude chirality--did you try including it and it was worse? 2. how many atoms and how many heavy atoms did the largest fragment in your library have, and how many connected rings did it have? 5. was the GraphAF model trained on very small molecules, e.g. QM9, or did you train it on a library of drug-like molecules? 7. I still don't understand what the "representation of explored chemical space" means here.  UMap is a technique for dimensionality reduction---what is the input to UMap?

---

> > > ### Author Response · Authors · 2021-11-24
> > > **Response to reviewer gg4D for additional clarifications**
> > >
> > > 1. Ablation study:
> > > The original random results doesn't include any part trained on data. The reason why our proposed architecture without training surpasses other baselines can come from two aspects. Firstly, we incorporate chemical knowledge to ensure only valid chemical edits. When constructing the fragment library, we decompose known molecules into fragments and the fragment attaching sites are recorded in each fragment as editable sites. When building new molecules, the adding and deleting sites are selected from the editable sites. Summarizing potential editable sites using known molecules can be considered as incorporating chemical rules into 3D molecular editing, which reduces the search space to only contain valid chemical edits. Comparing the performance of original random (allow editing based on chemical rules) and the new random (allow editing at any sites), space reduction can improve the Prod score from 0.29 to 0.46. Secondly, we treat the 3D molecule generation problem as an optimization task in 3D space through adaptive learning and simulated annealing. Without adaptive learning, the Prod score is 0.46, while with adaptive learning, the Prod score is 0.53. Previous 3D model, liGAN, learns 3D molecular embeddings from CrossDocked2020, which contain molecules that are not optimal for the target binding site. Therefore, it is hard to generate highly optimized molecules from this kind of embedding. Previous 2D models ignore the 3D structural information of the ligand and the target. Even though they optimize the score from data-driven activity predictors,  the generated molecules may not be considered optimal when evaluated using molecular docking, which emphasizes on the 3D interactions between the ligand and the target.
> > >
> > > 2. Method details: \
> > >   a. Chirality:  chirality is used to describe the relative 3D position of functional groups surrounding an atomic center. Therefore, calculating Morgan fingerprint with chirality is only meaningful for 3D molecules. To make a fair comparison between 2D and 3D molecular generation models, we decide to not include chirality in Morgan fingerprint. \
> > >   b. Fragments:  In our paper, we set the fragments to have no more than 10 heavy atoms and this setting can be changed for specific applications. We assume that "the largest fragment" referred by the reviewer means the fragment with the largest number of heavy atoms. In our current setting, there are 226 fragments with 10 heavy atoms. Their total number of atoms range from 12 to 28 and they all have two connected rings.
> > >   c. GraphAF:  We follow the original setting in GraphAF to train their model, which uses ZINC250 dataset[1] as the training set. ZINC250 is a commonly used resource for training drug design models, and molecules in ZINC250 have up to 60 heavy atoms[1].
> > >   d. UMap representation:  The input of UMap is CDDD[2] descriptors generated by the default model. With UMap dimension reduction, we can visualize the distribution of molecules, which we refer to as "representation of the explored chemical space". In Figure 4 A., we compare the distribution of generated molecules at step 10, 100, 1000, and among them we keep the position of molecules generated at step 10 (colored in black) still and use them as a reference for easy comparison. From Figure 4 A, we can tell that as the sampling chain proceeds, there is obvious molecular distribution shift, which corresponds to better objective score as demonstrated in Figure 4 B. We want to use this figure to illustrate the idea that by adaptive learning and simulated annealing the distribution of the generated molecules moves towards a more desired space.
> > >
> > > [1] Irwin, John J., et al. "ZINC: a free tool to discover chemistry for biology." Journal of chemical information and modeling 52.7 (2012): 1757-1768.
> > >
> > > [2] Winter, Robin, et al. "Learning continuous and data-driven molecular descriptors by translating equivalent chemical representations." Chemical science 10.6 (2019): 1692-1701.

---

### Official Review · Reviewer_X9Jm · 2021-11-01

**Correctness:** 3
**Technical Novelty And Significance:** 2
**Empirical Novelty And Significance:** 2
**Recommendation:** 5
**Confidence:** 4

**Main Review:**

The main idea of the paper is summarized in the above sentences. The advantages and the disadvantages of this paper from my side are as follows.

Strengths:
1. This paper mainly focuses on knowledge-guided geometric editing. As for the knowledge, as far as I read, including (point out if I am not correct): autodock vina (for target-ligand energy), QED (drug-likeness), SAscore (for synthetic accessibility). The knowledge is important and necessary for molecule design.
2. The geometric editing stands on the bonds in the molecule (between fragments), which causes stability and the specific property. The editing method is in a reasonable design, which selects the bond and the fragment, then generates a new molecule.
3. The results on the selected target perform well in the 3D generation w.r.t the evaluation metrics.

Weaknesses:
Though this paper gives knowledge-based editing, there are many concerns I have,
1. In the big picture, GEKO is a self-learning method based on the knowledge information provided by some softwares. Actually, the formulation of Equation (1) and Equation (16) is exactly the reinforcement learning-based method, or specifically, the policy gradient method. Equation (1) is indeed the reward function provided by the ground-truth (though provided by the software), and Equation (16) is the reward weighted objective. In this way, the claim of this paper that it is an unsupervised drug design is somehow overclaimed. Besides, the authors even do not talk about RL at all.
2. Detailed modeling questions. I think $x_{frag}$ is different from the $x_{skel}$, which should be the new fragments in the fragment library. However, the definition of the softmax equation (10) can not reflect at all, and where is $x_{frag}$ in equation (10)? What is the detailed softmax? Similarly, in equation (13), where is r in the right side?
3. I am confused about the edge r and the bond a, what is the difference? why there are two samples of r and a?
4. As for the four atoms selected u, w (skel and frag), how are they selected? Since each bond only has two atoms, and what are the two other atoms selected? Footnote 4 is missing.
5. In the overall formulation, the dimension of the formulation are not introduced, which makes feel confused about the transformation, for example, the $p_{attach}$.
6. The training cost is also important since this is a self-learning method and the training data is generated by the model itself. Therefore, it is important to report the cost and the comparison with other baseline methods.
7. More importantly, the following content of dihedral angle $\alpha$ is missing. What is then after sampling an angle $\alpha$?
8. Minor points, some typos are existed, for example, 'seleted' in Figure 1.


**Summary Of The Paper:**

This paper works on the deep learning models for automatic drug design. The authors propose a method named GEKO, the geometric editing under knowledge guidance, which incorporates the physicochemical knowledge into the designed 3D model. The authors claimed this is an unsupervised drug design method that the training is conducted in a purely training data free fashion. The training algorithm is based on geometric (3D) editing with the knowledge (Vina, QED, SA core) guidance by self-training and simulated annealing. The representation of the molecule is modeled at atom-level and fragment-level. With the geometric editing process continuing, the model continues updating and learning towards a better result. The method is evaluated on the drug design task on two sets of target datasets, by generating the molecules (drugs), the results are evaluated by multiple metrics. The performances are improved compared with other baseline methods (including both 2D and 3D methods).


**Summary Of The Review:**

The method contains several signs of progress to generate new drugs in 3D space. Overall it is ok, but more questions should be answered.

---

> ### Author Response · Authors · 2021-11-20
> **Response to Reviewer X9Jm**
>
> We thank Reviewer X9Jm for the helpful suggestions for improving our method details. Following the reviewer's suggestion, we clarify the method details in our paper. We also address these issues below.
> 1. Comparison with Policy Gradient Algorithms: Standard policy gradient algorithms[1] need to estimate the value function, i.e., the outcome from some state. However, in our sampling process, we cannot know its future outcome when collecting the data, since the outcome only shows after completion of the sampling sequence and we only do this sampling once for all chains. In fact, we design our algorithm based on simulated annealing and apply a self-learning technique on this process. Our method is more like a parameterized MCMC with annealing rather than a RL method.
> 2. x_frag is the graph of the selected fragment from the fragment library, which is determined jointly by x_skel and the selected editable bond r on x_skel. MLP_2 is a mapping from d-dim to |H|-dim, where H is the rigid library, so that each fragment in vocabulary gets a score, which is then fed into softmax function.
> 3. In "add" operation, we need to select a directed edge r in x_skel and a directed edge a in x_frag. The ending node of edge r and edge a are deleted and the beginning nodes are connected to form a new edge. We need to sample both r and a because the skeleton molecule and the selected fragment both have multiple sites that can be edited. This mechanism is described in Figure 1 and "Operations of Geometic Editing" section.
> 4. An dihedral angle is defined by 4 atoms, w', v', v, w, as illustrated in Figure 1 upper left panel. w and w' atoms are neighboring atoms of atom v and v' respectively. In our implementation, we predefine a neighboring atom w of v for edge (v, v') and a neighboring atom w' of v' for edge(v', v).
> 5. We supplement the necessary details including dimensions for readers to better understand the transformations.
> 6. Using JNK3 experiment as an example, GEKO takes 4 days 6 hours on 60 cores to sample 5000 parallel chains and 1000 steps, and the computation time includes geometric editing, objective calculation (Vina score, QED, SA) and model training. MARS takes 12 hours to converge, GA+D takes 2.2 hours, RationaleRL takes 5.7 hours to fine-tune, MolDQN takes 10.5 hours. It should be noted, molecular docking is a time consuming step in our model. Compared to the conventional drug design and optimization process, which takes months to years, the time spend on training our model can be ignorable.
> 7. After predicting the dihedral angle, the skeleton molecule and the fragment are connected using the selected bonds and the dihedral angle as described in "Operations of Geometric Editing" section.
>
> [1] Sutton, Richard S., and Andrew G. Barto. "Reinforcement learning: An introduction" MIT press, 2018.

---

> > ### Comment · Reviewer_X9Jm · 2021-11-23
> > **Clearer than before**
> >
> > Thanks for clarifying some details of the paper writing in the current version, the responses give me some clear understanding now. For example, the predefined neighbors. I still encourage the authors to better refine the paper writing.
> >
> > For the first question about reinforcement learning, it is not required to predict the value function. Actually, the intermediate reward or even delay reward can also be effective. Therefore, it is still very relevant to the RL-based training, and I still encourage the authors to discuss or cite related papers.
> >
> > I am interested in the paper and I think that if the authors can give the open-source code, it would be very helpful.
> >
> > I will increase the score and I look forward to other reviewers for better discussion.

---

> > > ### Author Response · Authors · 2021-11-24
> > > **Thanks for the response**
> > >
> > > We want to thank Reviewer X9Jm for the positive feedback and will release our code in the near future. We also appreciate it if all the reviewers can have more discussions regarding our paper or 3D drug design in general, as it is a relative new field but definitely worth to study.

---

### Official Review · Reviewer_ExEq · 2021-11-01

**Correctness:** 3
**Technical Novelty And Significance:** 2
**Empirical Novelty And Significance:** 3
**Recommendation:** 5
**Confidence:** 4

**Main Review:**

Strength:
1. The proposed model outperforms other baselines in the multi-objective molecules optimization benchmark.
2. This model doesn't rely on a data-driven biological activity predictor.


Weakness:
1. The model seems to be incremental. MARS already used rigid fragment library, editing operations, and MPNN. GEKO extends them to geometric editing, HMPNN, and docking software. At the first glance, I thought those rigid fragment library and editing operations are brand new concepts, but I found it's not after reading the MARS paper. It'd be nice if the authors contrasted GEKO and MARS so that readers would not be misled.
2. There are always pros and cons for each direction. The good thing about using a data-driven biological activity predictor is that binding site information is not required. Therefore, it doesn't require the 3D structure of the target, which makes it to be applicable to many proteins. However, when using docking software, data-driven predictors of biological activity are not required, but the 3D structure and binding site information are essential.
3. When running docking software, different poses of a molecule should be generated and tested. However, that information seems to be missing in the method section. In general, the pose generation is a big bottleneck, so if this information is included in the Methods section, it will help to better evaluate this paper.

**Summary Of The Paper:**

This paper proposes a method to optimize the multiple properties of a molecule. They first define rigid fragment library as a building block, which is used to generate a new molecule through geometric editing. They represent a molecule using HMPNN and the whole model is trained using the weighted maximum likelihood estimation. The proposed model is compared with some baselines on several multi-objective optimization tasks and shows good performance.


**Summary Of The Review:**

I have some questions/suggestions;
1. Optimizing speed comparison would be a good addition because docking simulation is known to be very slow.
2. In equation (15), the angle is sampled from a softmax funciton? Is the angle categorical?
3. In the baselines, it's known that VJTNN[1] is later model than JT-VAE. In addition, MolDQN[2] and CMG[3] are other recent models. Better to look at them.

[1] Jin, Wengong, et al. "Learning multimodal graph-to-graph translation for molecular optimization." arXiv preprint arXiv:1812.01070 (2018).

[2] Zhou, Zhenpeng, et al. "Optimization of molecules via deep reinforcement learning." Scientific Reports 9.1 (2019): 1-10.

[3] Shin, Bonggun, et al. "Controlled molecule generator for optimizing multiple chemical properties." Proceedings of the Conference on Health, Inference, and Learning. 2021.

---

> ### Author Response · Authors · 2021-11-20
> **Response to Reviewer ExEq**
>
> We thank Reviewer ExEq for the sincere suggestions. Here, we address your questions:
> 1. MARS and GEKO are editing-based methods. MARS generates 2D molecules and GEKO generates 3D ones. They have 3 differences:
>   a. MARS generates fragments by breaking single bonds. For a molecule with n single bonds, 2n fragments are generated and each fragment has one attaching point. GEKO breaks all single bonds together, which leads to n+1 fragments, and each one can have multiple attaching points. This distinction is tailored for their applications. MARS deals with 2D molecules, which has no structural flexibility. GEKO generates 3D molecules whose structural flexibility is caused by single bond rotation. If a fragment includes rotatable bonds, it corresponds to multiple conformers, all of which should be included in the library. To increase the coverage of our fragment library, fragments are made rigid by limiting rotatable single bonds.
>   b. MARS conducts editing as follows: decide "add" or "delete"; if "add", choose a node in the skeleton atomic graph and choose a fragment to add; if "delete", choose an edge in the skeleton atomic graph to delete. GEKO firstly decides "add" or "delete"; if "add", choose an addable edge in the skeleton fgraph, choose a fragment from the library, choose an attachable edge in the fragment and decide an dihedral angle to connect the skeleton and the fragment; if "delete", choose an deletable edge in the skeleton graph to delete. The addable and deletable edges in GEKO limit edits between fragments and maintain the structural integrity of each fragment, as a partial fragment may not stay in its original 3D conformation.
>   c. MARS uses a data-driven bioactivity estimator and GEKO uses molecular docking to predict bioactivity. MARS cannot be directly combined with molecular docking. Molecular docking includes two modes: global pose search and local optimization. 2D molecule generators can only predict 2D structures and require global searching to predict the 3D binding pose with the target, which is slow. GEKO can generate 3D target-ligand complexes and only needs a local energy minimization, which makes molecular docking 100x faster and feasible.
> 2.  Recent advances in structure biology, e.g. X-ray, NMR and cryo-EM, make solving protein structures experimentally easier reflected by the rapid increase of protein data[1]. Computational tools like AlphaFold [2] achieves experimental level accuracy in single protein structure prediction. These advances make 3D drug design more accessible in practical drug research. But data-driven bioactivity predictors rely on experimental data for a specific protein including both actives and inactives. For a protein, finding a highly active or even drug-like compound is very challenging, not to mention the labor, time and money spent on experiments. Considering the accessibility of required data, 3D models are more generalizable.
> 3. As aforementioned, the global pose search is slow. GEKO avoids it by learning how to generate 3D molecules in the binding site. Full docking including global search is only conducted during evaluation for both 2D and 3D models. We use RDKit to generate the initial poses for 2D models and Vina to dock them into the binding site. For 3D models, we use the model generated pose as input for Vina. Top 1 Vina score pose is selected for evaluations.
> 4. Dihedral angle prediction is treated as a classification task with 10 degree bins. This setting is commonly used in proteins structural prediction[3].
> 5. For JNK3 experiment, GEKO takes 4 days 6 hours on 60 cores to sample 5000 parallel chains and 1000 steps, and the computation time includes geometric editing, objective calculation (Vina score, QED, SA) and model training. MARS takes 12 hours to converge, GA+D takes 2.2 hours, RationaleRL takes 5.7 hours to fine-tune, MolDQN takes 10.5 hours. Molecular docking is a time consuming step in our model. Compared to conventional drug design, which takes months to years, the time spend on training our model can be ignorable.
> 6. VJTNN and CMG are designed for lead optimization, which starts from an existing active and optimize it. They are trained to map from one molecule to another one with better properties using paired data. GEKO focus on de novo drug design, which generates molecules from scratch. Also, there is no molecular pair data for the targets we evaluated, which makes VJTNN and CMG not comparable. We add MolDQN into our comparison.
> [1] Van Montfort, et. al. "Structure-based drug design: aiming for a perfect fit." Essays in biochemistry 61.5 (2017): 431-437.
> [2] Jumper, John, et al. "Highly accurate protein structure prediction with AlphaFold." Nature 596.7873 (2021): 583-589.
> [3] Gao, Jianzhao, et. al. "Grid-based prediction of torsion angle probabilities of protein backbone and its application to discrimination of protein intrinsic disorder regions and selection of model structures." BMC bioinformatics 19.1 (2018): 1-8.

---

> > ### Comment · Reviewer_ExEq · 2021-11-22
> > **Thanks for the Response**
> >
> > 1. Yes, this response looks better than the original manuscript, although the novelty is not that significant.
> > 2. You don't need to have all bioactivity pairs for an in-silico binding prediction. However, you do need to have all target 3d structures to apply your method. In addition to 3d structures, you also need to know the binding pocket location to apply your method. These two things are what I pointed out in the earlier review.
> > 6. How about just generating a new molecule from an existing one instead of from-scratch in order to be comparable? The reason why this is important is we can't say how good the proposed model is based on the small benchmarks you presented.

---

> > > ### Author Response · Authors · 2021-11-30
> > > **Response to the additional comments**
> > >
> > > * As mentioned in the previous response, solving protein structure and identifying ligand binding sites has become easier in recent years due to the rapid development of structural biology. In addition, computational methods (e.g. AlphaFold and Fpocket) can also be used to solve these problems. These advances make our method GEKO more accessible to all drug targets, as it only requires 1 target structure and the target binding site to apply. In comparison, 2D drug design methods require a large amount of active and inactive compounds for a specific target in order to train an accurate activity predictor. For example, the DRD2 activity predictor used in VJTNN and CMG is developed using 7218 actives and 100,000 inactives. Collecting this data requires intensive lab work, which is expensive and time-consuming,  and cannot be replaced with in-sillico approaches. Therefore, comparing the easy-accessibility of the required data for 2D baseline models and our proposed GEKO, we believe GEKO is more generalizable to different drug targets.
> > > * Following the reviewer's suggestion, we conduct a constrained optimization task using GEKO. We use DRD2 target as an example, which is the same target used by VJTNN and CMG. The task is defined as optimizing molecular properties from a known active molecule, and constraining the similarity between the initial molecule and the generated molecule. We ran a 1000 chain experiment using GEKO to simultaneously optimize Vina_min score, penalized logp, QED and sim(X,Y) (the similarity between the initial molecule X and the generated molecule Y). We only compare GEKO with VJTNN, since CMG doesn't provide sufficient information for reproducing its results (more specifically, its training data, model hyperparameters and instructions for running the code are missing). VJTNN is originally not designed for multi-objective optimization, and we use VJTNN to optimize the objectives sequentially following the baseline setting described in CMG paper. We evaluate the models using success rate, diversity, novelty following VJTNN. The success rate is defined as Vina_dock < -8.18, QED > 0.25, penalized logP is improved more than 1 compared to the initial molecule and sim(X,Y) > 0.4. The success thresholds remain the same as used in our paper except for the newly added objectives, penalized logP and sim(X,Y), whose success threshold is defined following CMG. From the results listed below, we can see that GEKO and VJTNN achieve comparable performances on success rate, but GEKO surpasses VJTNN on both novelty and diversity. GEKO and VJTNN's performances are relative low on the success rate, and this may be caused by the fact that simultaneously optimizing multiple properties while maintaining structural similarity is very challenging. Without major alterations on the molecular structure, the explorable space for optimizing all required objectives becomes highly sparse.
> > > |Method|Succes|Novelty |Diversity|
> > > |---|---|---|---|
> > > VJTNN|1.0%|40.0|0.859
> > > GEKO|1.0%|97.7|0.927

---

> > ### Author Response · Authors · 2021-11-24
> > **Follow up on the discussions**
> >
> > We want to take this opportunity to thank Reviewer ExEq again for the careful evaluation and sincere suggestions. We are willing to discuss more on this topic and answer any additional questions. Please feel free to let us know.

---

### Official Review · Reviewer_ZrRD · 2021-11-09

**Correctness:** 4
**Technical Novelty And Significance:** 2
**Empirical Novelty And Significance:** 3
**Recommendation:** 6
**Confidence:** 3

**Main Review:**

The approach described appears plausible. The methods are mostly well described, and the paper is clearly written for the most part.

Although the results are good, most of the proposals in the description of GEKO seem to be based on existing literature, and indeed described in the prior work section of this paper (e.g., MARS). So the paper seems less about a novel approach and more about SOTA performance. But the dataset used to test the model is a bit small if this is where the novelty of the paper lies. Only 12 targets is a very small dataset size when comparing model performance, and unlikely to be representative of the variation seen in a practical setting.

Have molecular docking software to estimate binding energy - but how accurate are these? In general I found the description of docking to be limited.

In the ablation study, ‘Random’ is chosen as a kind of baseline, but the score is better than expected if just randomly picking editable bonds without any deep learning. Why is this? In general more analysis of the ablation study findings would be helpful.

The authors claim GEKO outperforms other methods significantly. But the claim of statistical significance is not backed up with p values or confidence intervals for the comparison. The authors should provide evidence of a statistically significant improvement over other models, or adjust the wording.

The authors claim 1D/2D approaches limit models to targets with rich activity data. This is not entirely true, as really we want models to be able to generalize to new, previously unseen targets. With a large enough dataset, this is not impossible.

“Finding a drug molecule that can cure specific diseases is one of the most significant yet challenging task in human development” - it’s a very minor point, but the field of human development is unrelated to the field of drug discovery. The authors may want to say it is a major and significant challenge in the field of biology/medicine instead.

**Summary Of The Paper:**

The authors propose an approach for drug design that generates molecules by simulating adding or deleting parts of the molecules, and using graphnets to capture atom and fragment level information and construct new molecules. They use simulated annealing to ‘edit’ the 3D structures, and docking simulations, drug-likeness and synthesizability to provide information back into training. In this way, the authors claim the model can be trained on self generated data. The authors compare with multiple baselines on a test set of 12 targets, including the current SOTA model, and report improved performance.

**Summary Of The Review:**

While the results on the test set are good, I’m not convinced that the results alone here are novel enough for acceptance.

---

> ### Author Response · Authors · 2021-11-19
> **Response to Reviewer ZrRD**
>
> We thank reviewer ZrRD for the candid review, and hope our general response addresses your question regarding the novelty and contribution of GEKO. Here, we address your other questions:
> - Test set: We follow the SOTA 3D model liGAN to evaluate methods on 10 targets, also follow MARS to use 2 common targets for 2D drug design. Our test set is comparable to if not larger than the test set used in previous work.
> - Molecular docking: is widely used to estimate binding energy. Its structure-based scoring function is in charge of calculating the binding energy. The scoring function takes the 3D protein-ligand complex, and sum up pair-wise atomic interactions to estimate the energy. Therefore, its scoring function use the physical forces to estimate the binding energy, which makes it generalizable to all targets. In our implementation, we chose Autodock Vina as our binding energy estimator. The performance of Vina scoring function is evaluated in CASF benchmark[1-2]. Vina ranks 1st place on docking power and among the first quarter on scoring power. [3] Vina has been used in practical drug research to identify active compounds, such as [4-5]
> - Ablation study:
> |Method|Uniq|SR|Nov|Div|Prod|Median|Top 10|
> |---|---|---|---|---|---|---|---|
> |Random|100.0| 31.4 | 100.0 | 0.909 | 0.29 | -8.80 |-12.09|
> |HMPNN (random weight) |100|50.0|100|0.914|0.46|-8.91|-11.99|
> |MPNN|100|54.1|100|0.911|0.49|-9.26|-12.28|
> |HMPNN|100|58.2|100|0.911|0.53|-9.24|-12.50|
>
> The original "random" proposal represents random weights for HMPNN model. Therefore the prediction of edits are constrained by the HMPNN architecture which only allows editing on the fragment level.  To make it more clear, we changed the original ”random“ proposal to ”HMPNN (random weight)“, and added a random proposal which represents randomly editing at the atomic level. Random editing at atomic level shows a clear performance drop, which further confirm that our designed HMPNN model for fragment-based editing plays an important role. In addition, all the methods presented in Table 2 are powered by simulated annealing as part of our framework, which also contributes to their good performance.
> In addition, 2D baselines are designed to generate and optimize molecules in the 2D space and are not optimal for generating molecules 3D molecules that can fit into the target binding site. The 3D baseline, liGAN, learns molecular properties from PDBBind, which contains molecules that are not highly active. Our editing-based strategy has the potential to effectively explore valid 3D molecular space without any labeled data. The HMPNN model is used to expedite the optimization process, and even without adaptive learning, our designed procedure is able to generate molecules with desired properties. With adaptive learning, the performance can be further improved.
> - p values: We calculate p values for Prod score comparison. When compared to liGAN, p value < .001 using 12 targets, which indicates GEKO is significantly better. When compared to 2D baselines, only 2 targets are presented as 2D models are not generalizable. The p values are less than 0.02 for all 2D baselines except MARS. Comparing with MARS, the p value is 0.16. The performance of MARS varies on different targets (the Prod score of JNK3 is 0.34 and GSK3_beta is 0.04), and therefore, it is harder to accurately estimate the score distribution using a sample size 2. When comparing the Prod score on each target, GEKO achieves 0.50 and 0.54 on JNK3 and GSK3_beta respectively, while MARS achieves 0.34 and 0.04.
> - 1D/2D vs. 3D: the 1D/2D baseline models use data-driven bioactivity predictor trained by fitting ligand structural features to their activity data for a specific target. As these models ignore the protein information, they are target-specific and cannot generalize to other targets no matter how much data is presented. Therefore, they cannot be applied to newly discovered targets with a few or no activity data. Instead, molecular docking uses physical forces to predict binding, which are general to all targets.
>
> [1] Liu, Zhihai, et al. "Forging the basis for developing protein–ligand interaction scoring functions." Accounts of chemical research 50.2 (2017): 302-309.\
> [2] Li, Yan, et al. "Comparative assessment of scoring functions on an updated benchmark: 1. Compilation of the test set." Journal of chemical information and modeling 54.6 (2014): 1700-1716.\
> [3] Gaillard, Thomas. "Evaluation of AutoDock and AutoDock Vina on the CASF-2013 benchmark."Journal of chemical information and modeling, 58.8 (2018): 1697-1706.\
> [4] Sahoo, Maheswata, Sangeeta Daf Lingaraja Jena, and Satish Kumar. "Virtual screening for potential inhibitors of NS3 protein of Zika virus." Genomics & informatics 14.3 (2016): 104.\
> [5] Singh, Shilpi, Urmi Bajpai, and Andrew Michael Lynn. "Structure based virtual screening to identify inhibitors against MurE Enzyme of Mycobacterium tuberculosis using AutoDock Vina." Bioinformation 10.11 (2014): 697.

---

> > ### Author Response · Authors · 2021-11-24
> > **Follow up on the discussions**
> >
> > We want to thank Reviewer ZrRD for the suggestions and comments again, and we would love to discuss more about this paper and answer additional questions if there is any. Please feel free to let us know.

---

> > > ### Comment · Reviewer_ZrRD · 2021-11-24
> > > **Thanks for the response**
> > >
> > > Thank you for the additional clarification. The ablation study is much clearer now, and the revision resolves several of my previous concerns.
> > >
> > > I think the main concern is around novelty, and I would agree with the other reviewers that the novelty of the paper is still somewhat limited.
> > >
> > > However, taking the revision in mind I have increased the score I gave the paper.

---

### Author Response · Authors · 2021-11-17
**General Response to All Reviewers**

We thank all the reviewers for their sincere comments and suggestions. We also would like to take this opportunity to clarify several key points in our paper.

We aim to explore a subdivision of drug design task, 3D drug design or more formally structure-based drug design, which is a commonly used strategy in practical pharmaceutical research. The problem is defined as generating 3D molecules conditioned on a target structure. The current state-of-the-art model, liGAN, can only generate a small amount of molecules with better molecular docking score, an indicator of bioactivity, than the existing actives. A recent paper accepted by Neurips 2021 [1] also focuses on this task, but only shows moderate improvement comparing to liGAN in the molecular docking score, specifically [1] is lower than liGAN by only -0.1 kcal/mol in median Vina score. The median and average molecular docking score of both liGAN and [1] cannot surpass the cocrystalized reference molecule in the protein structure, which limits these methods to generate useful impact in practical applications. Our proposed method, GEKO, demonstrates a major breakthrough on this task by achieving a lower median Vina score than liGAN and the difference is -1.77 kcal/mol. Our results indicate GEKO is a more suitable method for 3D drug design task.

In addition to the state-of-the-art performance achieved by GEKO, we consider our algorithm design to be not only useful but also novel.
1) GEKO uses an editing approach to generate new molecules, and editing is a commonly used method in 1D/2D molecular generation (e.g. MARS, MolDQN, GA+D). However, how to effectively conduct editing in the 3D space is a new task that has not been studied by any previous work. Compared to 1D/2D molecules, the additional dimensions in 3D space introduces more degree of freedom in the molecule, which is proportional to its number of atoms. Directly predicting atomic position or interatomic distance has been used in 3D conformation prediction, however, these methods show obvious performance drops when applied to drug-like molecules due to their relative large molecular size. From a chemical perspective, the position of each atom in a molecule is constrained by chemical rules, and dihedral angle is the one controlling the conformational flexibility of a molecule. We propose to break dihedral angles to generate a rigid fragment library. Our model predicts the attaching sites on the skeleton molecule and the selected fragment and combines them using a predicted dihedral angle. In addition, our method preserves hydrogen atoms, which defines the orientation of the attached fragment, to guarantee chemical validity and reduce the complexibility of generating 3D molecules. Therefore, GEKO proposes an editing-based approach to model dihedral angles in order to generate new molecules, which is a novel method that has not been used before.
2) GEKO effectively extracts molecular docking knowledge to assist 3D drug design.  There is limited amount of experimental measured 3D target-ligand complexes, and molecular docking offers an alternative learning resource. liGAN and [1] learn molecular docking knowledge using CROSSDOCK2020, which is a molecular docking dataset generated based on PDBBind[2-3]. PDBBind contains ligands with a wide range of bioactivities[2] including those with relative low bioactivities. Learning from these molecules cannot guarantee drug-like bioactivities of the generated molecules. GEKO employs an adaptive learning approach to learn the molecular docking score change upon each edit. Using molecular structures generated by local edits along the Markov chains, more granular knowledge can be extracted from molecular docking. Our results also indicate learning molecular docking in the propose approach can achieve better model performance.

In summary, we consider GEKO as a novel approach to tackle the problem of 3D drug design. It utilizes an editing approach to model the dihedral angles between rigid fragments and adaptively learns the granular knowledge from molecular docking to assist 3D drug design. Based on these strategies it achieves state-of-the-art performance on 12 targets. More responses to each reviewer's comments will be posted soon.

[1] Luo, Shitong, et al. "A 3D Generative Model for Structure-Based Drug Design."  Thirty-Fifth Conference on Neural Information Processing Systems. 2021.

[2] Liu, Zhihai, et al. "PDB-wide collection of binding data: current status of the PDBbind database."
Bioinformatics  31.3 (2015): 405-412.

[3] Liu, Zhihai, et al. "Forging the basis for developing protein–ligand interaction scoring functions." Accounts of chemical research 50.2 (2017): 302-309.

---

### Decision · Program_Chairs · 2022-01-20

**Decision:**

Reject

**Comment:**

The authors describes a drug design method that generates molecules by simulating adding or deleting parts of the molecules, and using graphnets to capture atom and fragment level information and construct new molecules.  Simulated annealing is used to ‘edit’ the 3D structures, and docking simulations, drug-likeness and synthesizability are used to provide information back into training.  The authors compare with multiple baselines on a test set of 12 targets, including the current SOTA model, and report improved performance.

Strengths:

- The proposed model outperforms other baselines in the multi-objective molecules optimization benchmark.
- The model doesn't rely on a data-driven biological activity predictor.

Weaknesses:

- The reviewers point out that the model seems to be incremental with respect to previous work.
- The reviewers have concernts about the reproducibility of the work and find a lot of details lacking.

This is a borderline paper with a majority of reviewers voting for rejection. I recommend the authors to addrses the weaknesses above and resubmit to another venue.